# Experimental Characterization of A-AFiM, an Adaptable Assistive Device for Finger Motions

Jhon Freddy Rodríguez-León [1] , Eduardo Castillo-Castañeda [1] , José Felipe Aguilar-Pereyra [2] and Giuseppe Carbone [3,*]

[1] Centro de Investigación en Ciencia Aplicada y Tecnología Avanzada Unidad Querétaro, Mecatrónica, Instituto Politécnico Nacional, Querétaro 76090, Mexico; jrodriguezl1600@alumno.ipn.mx (J.F.R.-L.); ecastilloca@ipn.mx (E.C.-C.)

[2] Automatización y Control, Universidad Tecnológica de Querétaro, Querétaro 76148, Mexico; faguilar@uteq.edu.mx

[3] Department of Mechanical Engineering, Energy Engineering and Management, University of Calabria, 87036 Rende, Italy

\* Correspondence: giuseppe.carbone@unical.it

**Abstract:** Robot rehabilitation devices are attracting significant research interest, aiming at developing viable solutions for increasing the patient's quality of life and enhancing clinician's therapies. This paper outlines the design and implementation of a low-cost robotic system that can assist finger motion rehabilitation by controlling and adapting both the position and velocity of fingers to the users' needs. The proposed device consists of four slider-crank mechanisms. Each slider-crank is fixed and moves one finger (from the index to the little finger). The finger motion is adjusted through the regulation of a single link length of the mechanism. The trajectory that is generated corresponds to the natural flexion and extension trajectory of each finger. The functionality of this mechanism is validated by experimental image processing. Experimental validation is performed through tests on healthy subjects to demonstrate the feasibility and user-friendliness of the proposed solution.

**Keywords:** robot rehabilitation devices; control; four slider-crank; image processing; healthy

## 1. Introduction

Rehabilitation robotics can be considered a specific multidisciplinary field involving human-robot interaction. In this field, clinicians, therapists, and engineers collaborate to help rehabilitate patients. Robot rehabilitation devices are attracting significant research interest, aimed at developing implementable technologies that can be easily used by patients, therapists, and clinicians. They are not intended to replace therapists, but they can be a valuable means to enhance the effectiveness of clinicians' therapies and increase the recovery rates and quality of life of patients. Furthermore, rehabilitation robotics allows self-treatment with remote supervision of several patients by a single therapist [1,2]. The hands offer autonomy to people's lives by providing physical interaction and grasping capabilities [3]. According to [3], developing countries are home to 80 percent of disabled people. This is because they are poor countries and have difficulty accessing rehabilitation services. Therefore, these countries need low-cost robotic devices that can help them in the rehabilitation process, which could help people with disabilities, as also stated in [3].

Rehabilitation therapies of the limbs of the body, especially in the fingers of the hand, are performed through passive and active movements. Therefore, robotic devices that allow assistance in rehabilitation processes must have the ability to adapt to the type of therapy. Many robotic devices are limited to certain movements due to the restriction of their degrees of Freedom (DoF) [1–3]. The rehabilitation systems of the fingers of the hand are composed of a hardware system, which is composed of the architecture of the robot, the actuation, the system of transmission, and the types of sensors that are used for the control

of movement. In addition, these devices may have one or more points of contact with the limb to rehabilitate, that is, the End Effector (EE) or exoskeleton [4–8].

There are currently several robotic devices that help in the process of rehabilitation of the fingers of human hands. Some of these devices use pneumatic actuation [3], and they also have the advantage of providing information on the progress of the patient during rehabilitation therapies. However, these devices have the disadvantage that they must have a compressor for their operation, as presented in [3]. Cable-operated rehabilitation devices, as presented in [9,10], allow the movement of the five fingers of the hand with a single actuator using a clutch system. However, it is limited to the adaptability of users. Currently, there are commercial robotic devices that allow assistance in rehabilitation therapies. One of them is the Amadeo System presented in [11,12], which allows the movement of the fingers individually, generating a semicircular trajectory. In addition, it provides three modes of action: active, completely passive, or caring.

In references [13,14], the authors proposed a design of a lightweight forearm exoskeleton for fine-motion rehabilitation using a slider crack and four-bar mechanism with more links and an actuator. Other authors, as in reference [15] using a rigid-soft combined mechanism for a hand exoskeleton, proposed a highly compact device. In reference to [16] a numerical and experimental validation of ExoFing (a "DoF finger mechanism exoskeleton"), they are investigated by focusing on the kinematic model and numerical simulations. A novel mechatronic exoskeleton for finger rehabilitation with sensors for detection motions and control is proposed in [17], the configuration of the exoskeleton can be fully reconstructed using measurements from three angular positions of sensors placed on the kinematic structure. In reference [18], a novel index exoskeleton with three motors to help post-stroke patients perform finger and training exercises is proposed. Moreover, in reference to [19], a novel underactuated finger exoskeleton to assist grasping tasks for the elderly with worn muscles strength has been designed. The weight of the wearable part of the proposed exoskeleton is 127 g, and the overall weight is 476 g. However, the existing solutions have some limitations in terms of the complexity of the control architecture as well as in terms of user-friendliness, which can be addressed with the proposed design solution.

This paper proposed an experimental characterization of a low-cost user-friendly device named A-AFiM (Adaptable Assistive Device for Finger Motions). This device can support the rehabilitation therapies for four fingers during flexion and extension motions. It is important to note that the full motion of a finger requires at least three degrees of freedom. However, the specific rehabilitation task does not require fully replicating all the feasible motions of a finger. Instead, a single repetitive finger motion path is necessary and sufficient to provide properly assisted finger exercise. Accordingly, the proposed device is based on a 1-DoF mechanism that can easily fulfill the required finger motions just by adapting one link length with beneficial features in terms of cost and ease of use. It is also to be noted that initial setup adjustments can be easily made to the link lengths of the proposed mechanism in case the therapist wishes to exercise the finger along a different path. The proposed design has innovative aspects in terms of low-cost and user-friendly features. It includes a novel EE that allows easy adaptation and user comfort. Moreover, the specific design prevents risks of impacts between the device and the finger with clear advantages from a safety viewpoint. The proposed novel device can be easily adjusted to different patients and different exercising protocols. The portability of the device allows for self-treatment at home. The new EE, user-friendly interface, and easy configuration capabilities allow patients to use the proposed device autonomously at home, thus enabling rehabilitation training without requiring continuous assistance by a physiotherapist.

## 2. Requirements for a Low-Cost Device

According to [20], the range of motion in a joint is measurable, which allows knowing and evaluating the progress in the rehabilitation processes. This data is used by clinicians. Currently, metacarpophalangeal joints (MCP), proximal interphalangeal joints (PIP), and

distal interphalangeal joints (DIP) are measured. These joints allow generating the flexion-extension movements, which are essential in rehabilitation therapies, as in the reference. Research articles on rehabilitation therapies show that lost movements of injured limbs can be recovered by repeated movements in the joints [21,22]. These movements are divided according to the concerned joint. Thus, we have fingers flexion and extension, as shown in Figure 1. In the references [5–23], several types of robots help in rehabilitation processes, this is because the mechanisms can generate controlled movement in the joints. The rehabilitation process allows people to regain control of their limbs after an illness or traumatic event. For example, in strokes, it is necessary to generate repetitive physical exercises such as flexion and extension movements of the fingers [24,25].

The values of the flexion and extension motion vary from one joint to another and from one human to another. Table 1 provides an average of typical ranges of motion values for the MCP, PIP, and DIP joints of the fingers, for a healthy human adult. Data have been obtained following the experimental procedure similar to the one that has been reported in [26]. In particular, in [26] we previously did experimental measurements on human hand sizes by considering five male/female humans. In this work, we further elaborated the experimental measurements on human hand sizes by enlarging the data set with five more measured hand sizes to get more statistical relevant average values. Furthermore, we have compared our data with other literature sources such as [27–29], which confirm the fitting of our data with average human hand sizes.

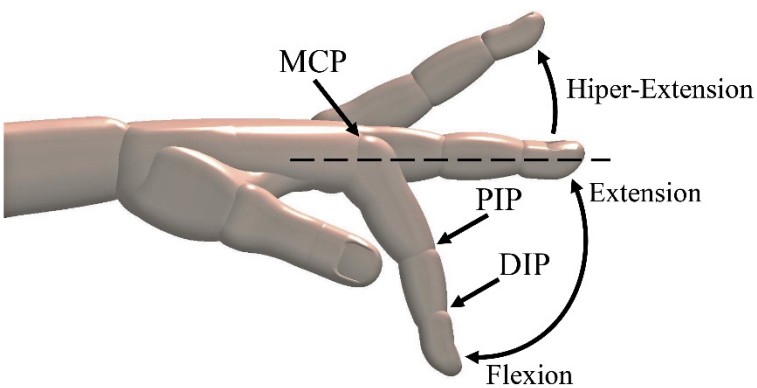

**Figure 1.** Fingers movements of flexion and extension.

**Table 1.** Fingers joints movements of flexion and extension.

| Fingers | Fingers Joint Movements (Degree) | | | | | |
|---|---|---|---|---|---|---|
| | MCP | | PIP | | DIP | |
| | Flexion | Extension | Flexion | Extension | Flexion | Extension |
| Pinkie | 90 | −32 | 106.5 | 3.5 | 74 | 5.5 |
| Ring | 89 | −26.5 | 107.5 | 6 | 72.5 | 5.5 |
| Middle | 90 | −26 | 106.5 | 6 | 75 | 5.5 |
| Index | 86.5 | −26 | 105.5 | 6 | 71.5 | 5.5 |

Index finger movement is analyzed for trajectory characterization. An orange circle was attached to the fingertip on a green background, controlled motions of flexion, and extension were generated, and the video was recorded (24 frames per second). A total of 450 images were acquired; for each image $x$, $y$ coordinates of the fingertip (point $Q$) are extracted by image analysis. Four images resulting from video decomposition are shown in Figure 2.

A color-based segmentation process was performed on these images for the generation of trajectory $Q$. This is done by using the ImageJ software [28,30], which allows knowing the RGB values of the point of interest. Through image processing, marker detection was used to identify the point $Q$ position; this strategy was explained in references [31,32]. The image

pixels are converted to mm. The generated trajectory is shown in Figure 3a. Figure 3b shows the trajectory of four fingers: index, middle, ring, and little. Those trajectories and the corresponding workspace were considered in designing a mechanical device for the flexion-extension of fingers.

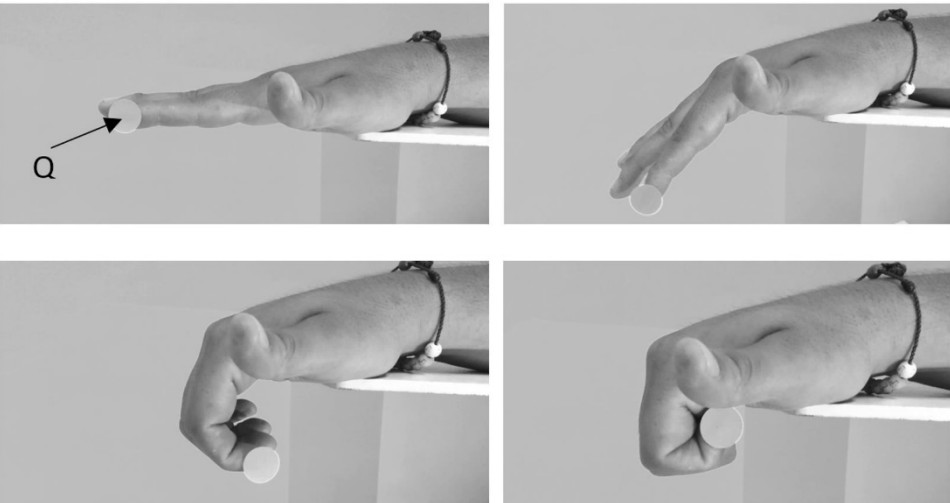

**Figure 2.** Decomposition of the video in images.

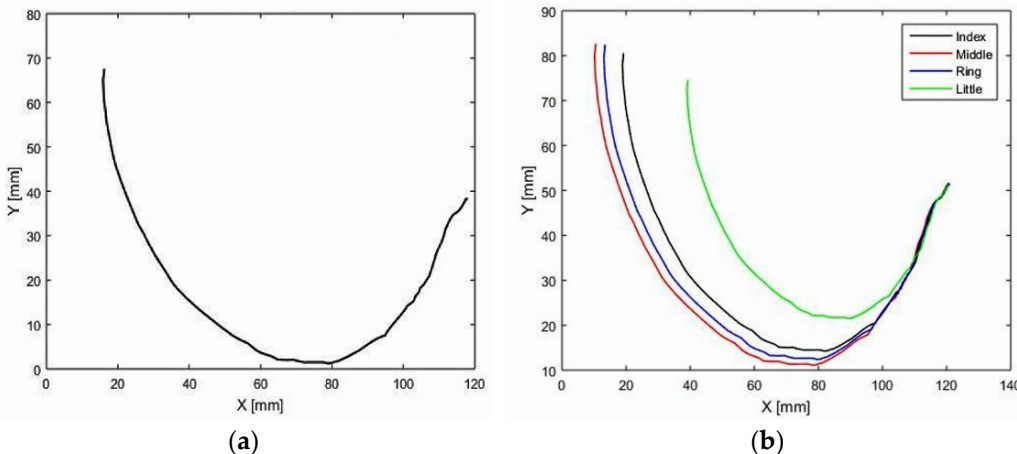

**Figure 3.** Trajectories of the fingers: (**a**) *x* vs. *y* trajectory of *Q* point (index fingertip); and (**b**) trajectories of the fingers Index, middle, ring, and little.

## 3. Conceptual Design

The length of each finger defines the dimensions of the corresponding mechanism. The synthesis of flexor and extensor mechanisms was performed from a trajectory generation method using three precision points based on the curves of the coupler link [33]. The Rotation-Rotation, Rotation, and Translation (R-RRT), and type slider-crank mechanisms are selected since the extension of the coupler link develops a very similar trajectory to that presented in Figure 3 [34]. The range of movement of the R-RRT mechanism is mechanically limited to avoid finger hyper-extension. The mechanism allows controlling the precision in the tracking of the trajectory since continuous and smooth movements are desired from the beginning to the end of the trajectory. Therefore, the position and velocity of movements should be controlled. The therapies generated by this mechanism can be configured by gentle and controlled movements. Thus, you can have various types of velocity and ranges of motion during bending and extension therapy. Therapies must always be supervised by professional physiotherapists to avoid sudden movements that can cause pain or further damage to the patient [35].

Figure 4 shows the corresponding CAD design for the index finger. The Direct Current (DC) servomotor generates a rotating movement that allows the linear displacement of point *A* in the slide. This movement is transmitted in the mechanism to point *E* where the semi-circular trajectory is generated. Point *E* is attached to the fingertip through a thimble secured by adhesive tape [35]. The EE is a passive articulation that allows the rotation and natural orientation of the distal phalanx. A safety system (SS) has been included by using a sensor that detects when the mechanism reaches its maximum position in bending or extension movements. As each user's fingers on their hand are different sizes, the mechanism can be reconfigured to adapt and generate the trajectory. A mechanism was proposed to allow reconfiguration of the mechanism and to modify the trajectory of point *E*. The average hand size for adults is presented in [36].

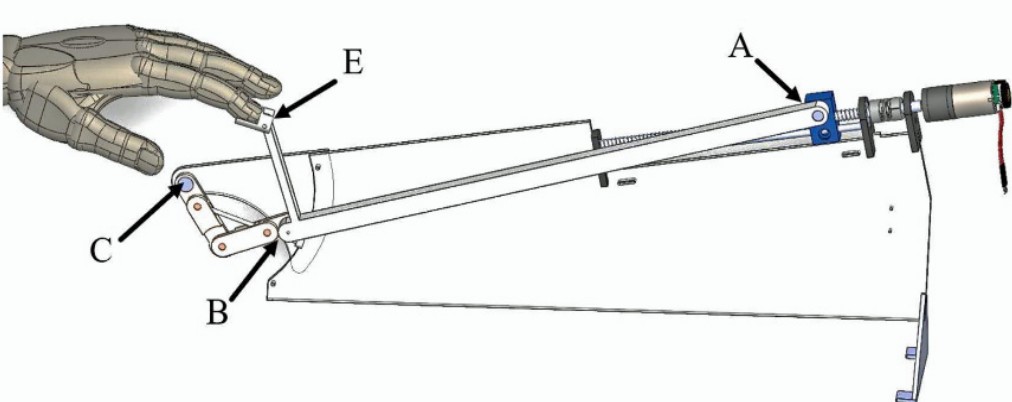

**Figure 4.** R-RRT mechanism of the index finger.

The mechanism can be reconfigured using variable *C*, proposed in Figure 5, whose length $L_1'$ between its external nodes *A* and *B* varies in the proportion of length, $L_4$ [34]. The *C* is composed of a passive rotational joint, point *F*, and a screw whose end is attached to the *G* point of link 5 through a ball joint [34–37]. The point of contact between the screw and link 4 is achieved through a nut; that $L_4$ varies when the screw turns. Similar triangles are formed by points *EFG*, and *AFB*, and they modify the angle $\delta$ as reported in [33–37].

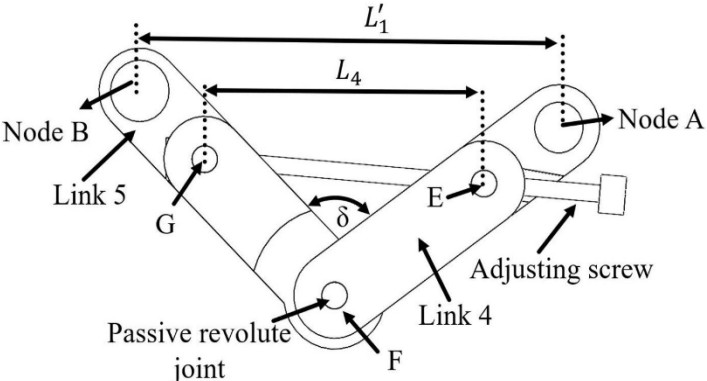

**Figure 5.** The variable-length crank C.

The proposed device consists of four mechanisms, type slider-crank to generate fingertip trajectories from index to little, as shown in Figure 6. Each mechanism is driven by a servomotor with only one active joint. It can reproduce the trajectory generated by the tip of each of the long fingers of the hand. When compared to other rehabilitation mechanisms on the market, its main advantages are ease of movement and storage, low cost of manufacturing and maintenance, and its lightweight structure. The mechanism

is composed of a rigid fixed structure and a mobile platform as an EE that guides the fingertips during the rehabilitation exercises.

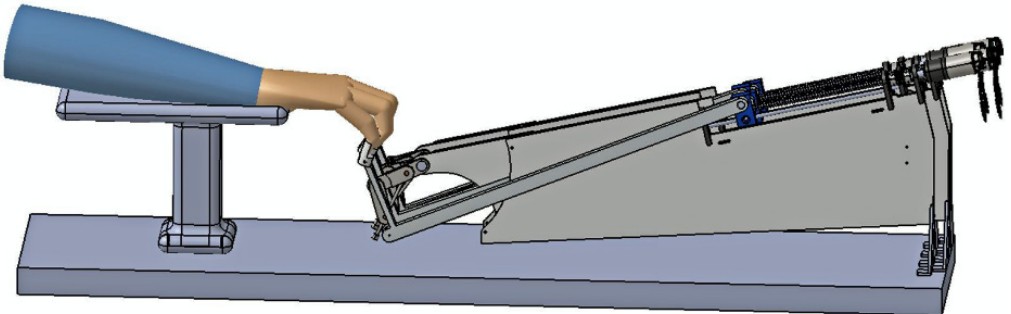

**Figure 6.** The four-bar slider-crank type mechanism.

## 4. Simulations and Prototype Development

The kinematic analysis of the slider-crank mechanism was performed applying the analytical method of complex numbers proposed in [37]. Figure 7 shows the vectorial representation of the mechanism, where **a**, **b**, and **c** are the vectors forming the kinematic triangle. The vector **c** corresponds to slider displacement restricted by limits of flexion and extension generated by the mechanism which represents the input of the system. With these data, the positions of the vectors are obtained.

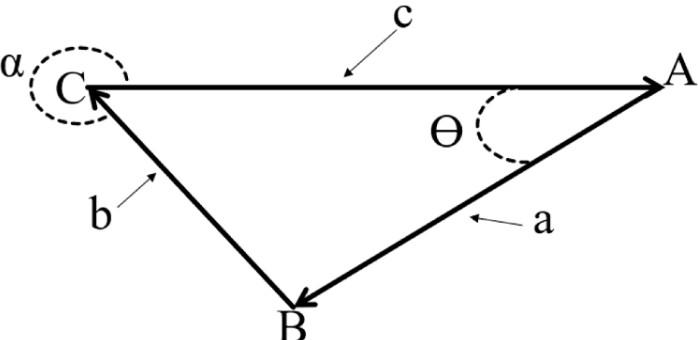

**Figure 7.** Vectorial representation of the mechanism.

In Figure 7. Considering that the values of the vectors **a**, **b**, and **c** are known. The parameters of the mechanism are calculated as

$$\mathbf{c} = \mathbf{a}e^{i\theta} + \mathbf{b}e^{i\alpha} \tag{1}$$

$$\mathbf{c} = \mathbf{a}e^{-i\theta} + \mathbf{b}e^{-i\alpha} \quad \text{complex conjugate} \tag{2}$$

Solve to find the value of $\theta$ and $\alpha$, therefore, it is possible to write

$$\mathbf{b}^2 = \mathbf{c}^2 + \mathbf{a}^2 - \mathbf{a}\mathbf{c}e^{-i\theta} - \mathbf{a}\mathbf{c}e^{i\theta} \tag{3}$$

$$\mathbf{b}^2 = \mathbf{c}^2 + \mathbf{a}^2 - \mathbf{a}\mathbf{c}\,cos(\theta) + \mathbf{a}\mathbf{c}\,isen(\theta) - \mathbf{a}\mathbf{c}\,cos(\theta) - \mathbf{a}\mathbf{c}\,isen(\theta) \tag{4}$$

$$\mathbf{b}^2 = \mathbf{c}^2 + \mathbf{a}^2 - 2\mathbf{a}\mathbf{c}\,cos(\theta) \tag{5}$$

$$\theta = arc\,cos\left[\frac{\mathbf{b}^2 - \mathbf{c}^2 - \mathbf{a}^2}{-2\mathbf{a}\mathbf{c}}\right] \tag{6}$$

$$\alpha = arc\,cos\left[\frac{\mathbf{a}^2 - \mathbf{c}^2 - \mathbf{b}^2}{-2\mathbf{b}\mathbf{c}}\right] - 360 \tag{7}$$

After checking the analytical method of the proposed design mechanism, a simulation has been programmed to validate the trajectory using the previous equations. Figure 8a shows the scheme of the vector representing this mechanism, and Figure 8b shows the results of the simulation.

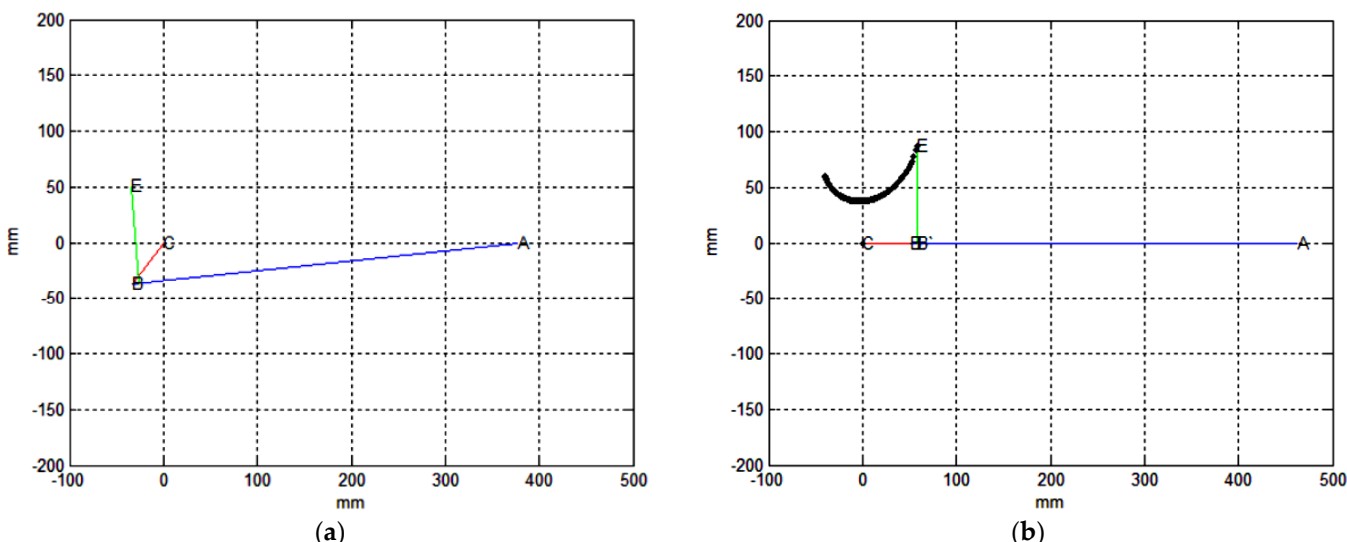

**Figure 8.** A design solution for the mechanical design: (**a**) A scheme of a design solution for the A-AFiM, and (**b**) simulation of the Flexion and extension trajectory.

The data obtained from the simulation represents a mechanism that is in the plane without any inclination. Therefore, to obtain the real data, the 0.3490 Radian mechanism must be tilted. This inclination is made by applying the rotation matrix to tilt the trajectory. The rotation matrix in on the *z*-axis is given by

$$Matrix_R = [cos(\lambda) - sin(\lambda)0 \, ; \, sin(\lambda)cos(\lambda)0 \, ; \, 0\,0\,1] \tag{8}$$

where $\lambda$ is the angle of rotation. Applying the previous rotation matrix to the data obtained in the simulation, the following graph is generated where the calculated trajectory and the trajectory are observed by applying the rotation to the *z*-axis. Power consumption and motor torque analysis have been carried out to analyze the expected performance of the proposed device as well as to properly select the motors for the proposed A-AFiM mechanism. For this purpose, the operation of the proposed device has been considered during flexion and extension exercises. The proposed simulation considers the average force of a human finger at $Ff = 1$ N, $Ff = 3$ N, and $Ff = 5$ N [24–27] having been applied to the EE. The simulation also considers the contribution of gravity. In Figure 9a, a picture of the maximum power consumption is shown, whereas in Figure 9b a maximum motor torque of the A-AFiM device is shown, including the proposed force in EE.

The control system includes an interface, four DC motors, two RoboClaw motor controllers [38], and an FTDI (Future Technology Devices International) device driver. To have satisfactory performance in a tracking position, motors should move in such a way to accurately provide the required kinematic position. The control implemented in the mechanism was performed by the experimental tuning method. The scheme connection is composed of two RoboClaw cards, on which each card can control two motors and make the acquisition of data generated by the two encoders in real-time. To send the controller data to the interface over a single channel, the cards were configured in the serial packet mode and a specific address was assigned to each card. In addition, the connection was made between *S1* and *S2* outputs according to the communication ports of each of the cards. These communication channels are connected to an interface converter, FTDI232, which

sends the data to the computer by the UART protocol and USB hardware. The schematic of this connection is shown in Figure 10.

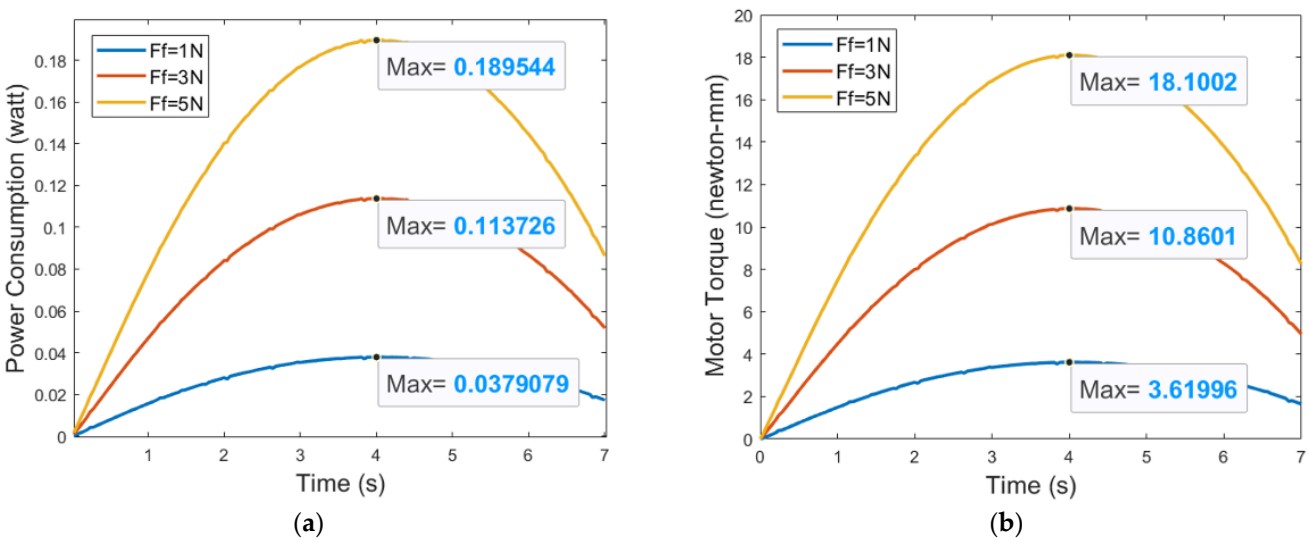

(a)

(b)

**Figure 9.** A-AFiM performance simulation: (**a**) Power consumption, and (**b**) Motor torque.

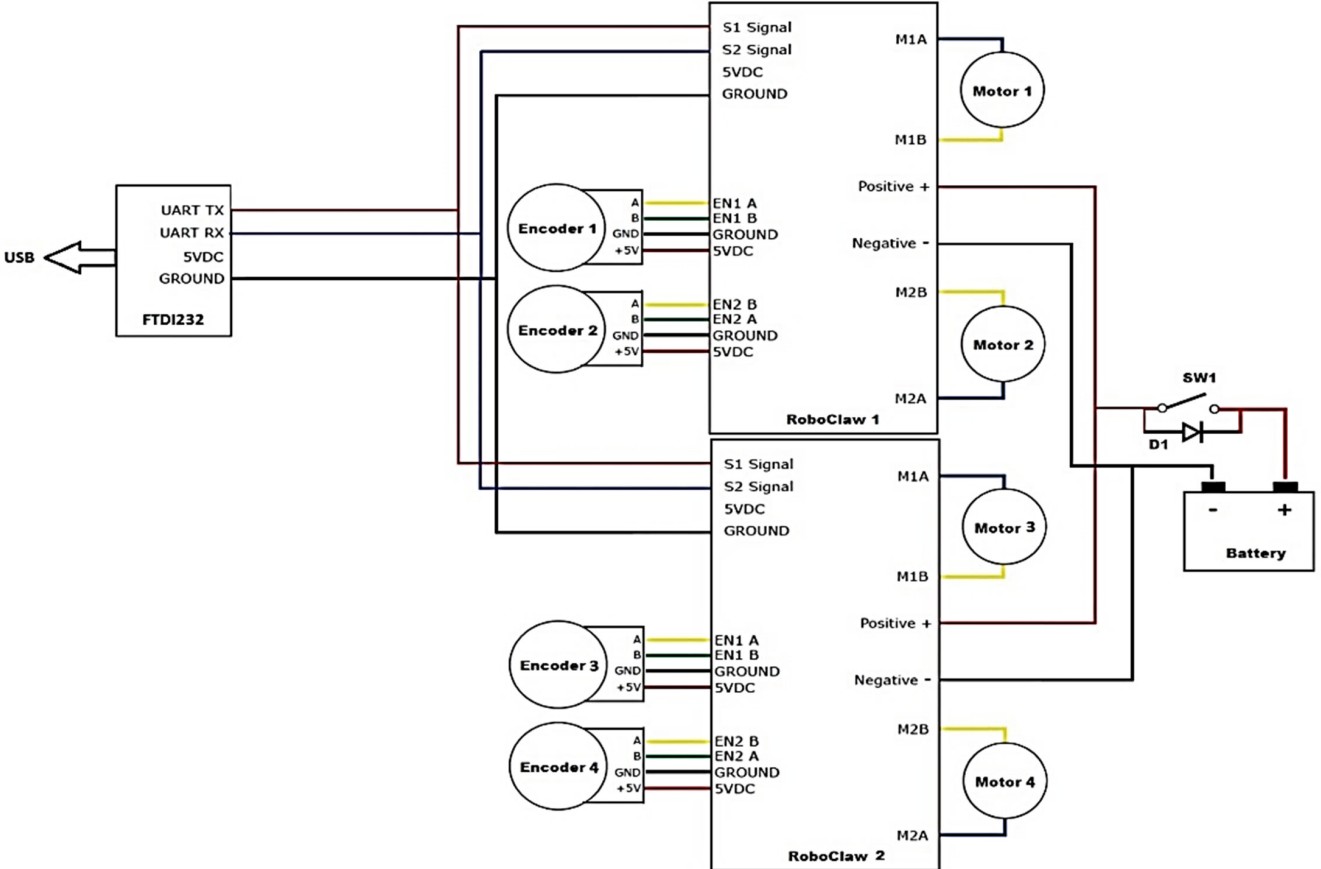

**Figure 10.** A scheme of the proposed control interface.

The control interface was designed in LabVIEW with specific characteristics. The interface was developed to control the motors independently with different input signals. A proper user interface has been defined according to Figure 11, which allows for path planning through PID control and the control through the pre-defined precision points

for the main rehabilitation exercises. The control interface was developed to generate coordinated movements with the following characteristics: motor control with a single input signal, synchronous movement, different trajectories, three types of speed programs (low, medium, and high), a single start button, and an emergency stop for the whole system. A simple closed-loop PID control is implemented to regulate the motor speed by using the feedback of an incremental encoder. Motor speed is an important parameter as it directly affects the speed of finger exercise. Accordingly, this parameter can be monitored in real-time. The user interface, Figure 11, includes a control for changing the motor speed according to the user's needs.

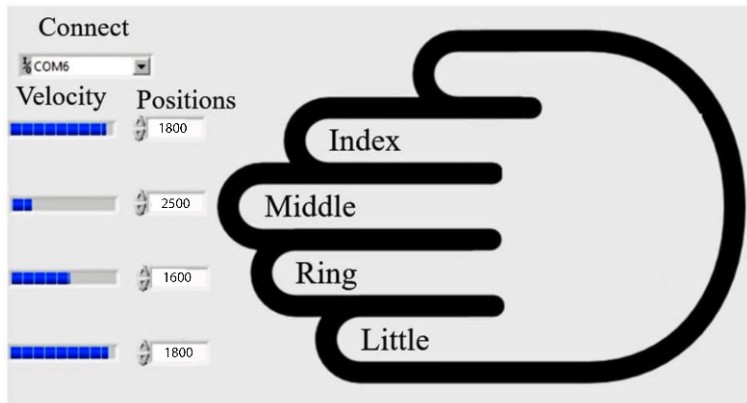

**Figure 11.** The developed graphical user interface.

After the design and simulation of an adaptable assistive device for finger motions, a prototype was produced, which was made up of a control interface, a controller system, four mechanisms, and a base for the hand, as shown in Figure 12. The design of the base supports the fingers of the hand, which can move in two axes, as shown in Figure 12. The objective of designing and building a base with 2-DoF allows for placing the fingers of the hand inside the workspace. Therefore, it allows moving different lengths of fingers on the hand to relate to the EE of each of the mechanisms. Consequently, the displacement that is generated along the *x*-axis allows the coupling of different-sized fingers of the hand with the mechanism, and the displacement that is generated along the *y*-axis allows the location of the movements of the fingers of the hand inside the working space of the mechanism.

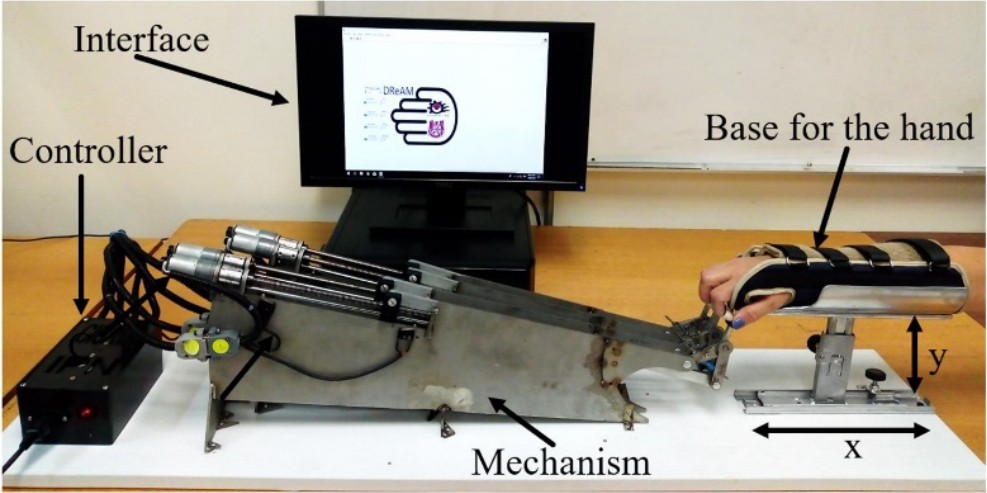

**Figure 12.** A-AFiM prototype.

## 5. Results and Discussion

The validation of the proposed prototype was made through the identification of trajectories proposed in [39]. The first test was to program the flexion and extension exercises to the minimum and maximum length of the crank for the workspace of the mechanical device. Therefore, the video was captured, and the images were extracted for analysis. Figure 13 shows the workspace and the trajectory of the index finger.

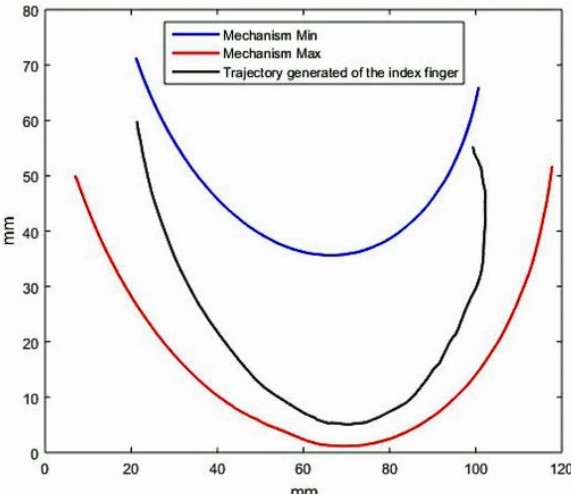

**Figure 13.** Comparison of the workspace and the index finger trajectory.

Figure 13 shows that the trajectory generated by the index finger in a natural way for the movements of flexion-extension is within the workspace of the mechanism in its entirety. The second test involves the acquisition of the index finger trajectory for seven people. All participants signed their informed consent forms to participate in the tests. The objective is to verify that A-AFiM can generate different trajectory types for different index finger sizes. Table 2 presents the characteristics of people's fingers. The trajectories obtained from each one of the index fingers are plotted within the A-AFiM workspace, shown in Figure 14.

**Table 2.** The features of people's fingers.

| N° Person | Gender [Male/Female] | Ages [Years] | Index Finger Phalanges [mm] | | | Weight [Kg] |
|---|---|---|---|---|---|---|
| | | | Distal | Middle | Proximal | |
| 1 | Male | 28 | 25 | 32 | 55 | 85 |
| 2 | Female | 21 | 22 | 30 | 52 | 55 |
| 3 | Male | 26 | 24 | 31 | 51 | 80 |
| 4 | Female | 22 | 21 | 30 | 51 | 60 |
| 5 | Female | 48 | 24 | 31 | 53 | 72 |
| 6 | Male | 62 | 25 | 31 | 54 | 89 |
| 7 | Male | 32 | 23 | 31 | 54 | 75 |

The previous figure shows that the trajectories of the index fingers are outside the A-AFiM workspace. Therefore, it makes use of the mobile base with 2-DoF, which can allocate the fingers of the hand within the workspace of A-AFiM. In Figure 15, the same trajectories of the seven index fingers are observed within the A-AFiM workspace due to the adjustment provided by the mobile base.

Considering the results obtained in Figure 15, it was possible to validate that A-AFiM can generate the movements of flexion-extension of the index fingers for people between 18 and 70 years of age, regardless of gender, age, weight, and size of the phalanges. The result obtained is based on the adjustment provided by the mobile base with 2-GDL. The third test suggested the validation of A-AFiM trajectories for long-fingered hands (Index,

Middle, Ring, and Little). Therefore, based on the data shown in Figure 3b, the dimensions of the long fingers and trajectories generated by the long fingers of the hand. It is possible to graph these trajectories within the A-AFiM workspace. The results obtained are shown in Figure 16.

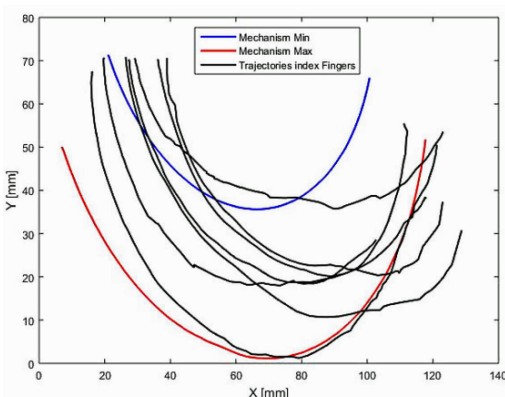

**Figure 14.** Workspace trajectories and index fingers within the A-AFiM.

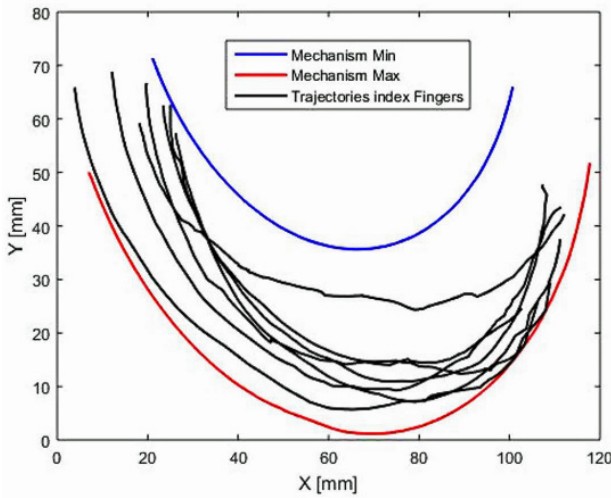

**Figure 15.** Workspace trajectories and index fingers.

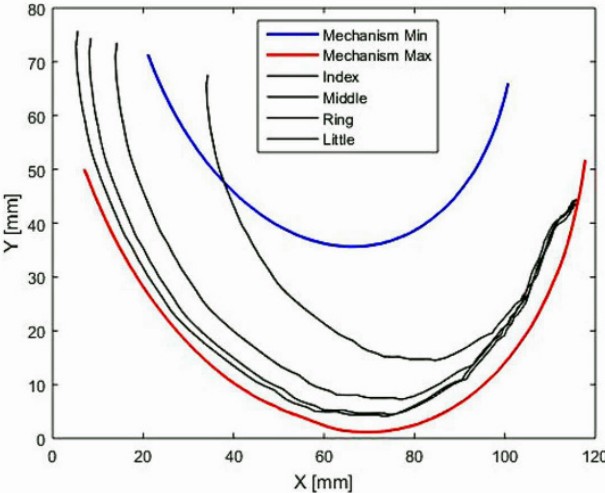

**Figure 16.** Long fingers hand trajectory validation graph.

Figure 16 shows that the trajectories of the index, middle, and ring fingers were entirely within the A-AFiM workspace. However, the path generated by the little finger was not entirely within the A-AFiM workspace. This problem was present since the coupling link of the mechanism corresponding to the little finger was generating an elliptical trajectory larger than that of the pinkie finger. Therefore, to solve this problem, it was necessary to resize the coupling link so that it could be adjusted to the size of the little finger. The last test was performed with three healthy people. The objective was to validate the functioning of A-AFiM through interaction with people. The results were based on the experiences expressed by users, who refer that: the system generates smooth movements, most of them prefer the average speed and feel comfortable inserting their fingers into the base that holds the hand, as it gives them the security to have a fixed hand.

The first snapshot, Figure 17a, refers to the beginning of the movement. Snapshot Figure 17b was taken in the extension motion, and snapshot Figure 17c was taken in the flexion motion. It is expected that the four long fingers should move at the same time and speed within the movements of flexion-extension, as this provides coordination between the different phalanges of the fingers, generating smooth movements while offering safety for users.

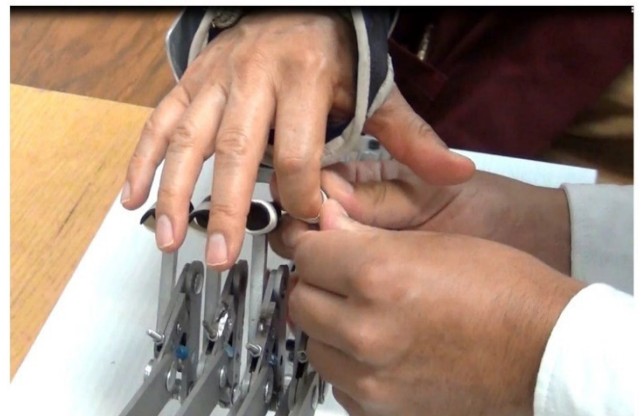

(**a**)

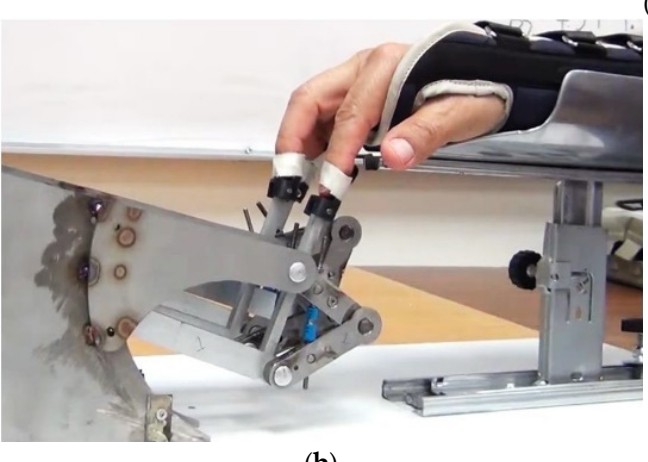

(**b**)

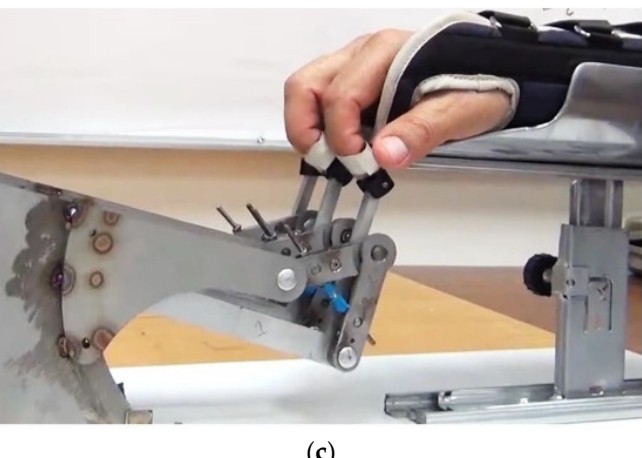

(**c**)

**Figure 17.** Snapshots of the A-AFiM motion during the test: (**a**) Starting position, (**b**) extension motion, and (**c**) flexion motion.

Preliminary tests have been carried out to estimate the performance and trajectory validation of the system in the flexion and extension rehabilitation motions when using strategy control and image processing. In future applications, the manufacturing and assembly tolerances of the components of the prototype can be significantly improved by using a high-precision manufacturing process.

## 6. Conclusions

This paper describes the design and validation of the A-AFiM, an Adaptable Assistive Device for Finger Motions. The main features of this device are simulated by considering its main slider-crank mechanism. Its workspace has been numerically calculated and compared with the required finger rehabilitation exercises for flexion-extension motions. A specific EE design and a control strategy have been proposed to generate suitable and user-tailored flexion-extension motions of the fingertips. Experimental tests have been carried out to validate the feasibility and effectiveness of the proposed device. Tests have been carried out on healthy people. The results of experimental tests have shown a good match and suitable operation of the proposed A-AFiM robotic device for the planned assistive motions.

**Author Contributions:** Conceptualization, J.F.R.-L., J.F.A.-P. and E.C.-C.; methodology, G.C. and E.C.-C.; software, J.F.R.-L. and E.C.-C.; validation, J.F.R.-L., J.F.A.-P. and E.C.-C.; formal analysis, E.C.-C. and G.C.; investigation, J.F.R.-L., G.C., J.F.A.-P. and E.C.-C.; resources, J.F.A.-P. and E.C.-C.; data curation, G.C., J.F.A.-P. and E.C.-C.; writing—original draft preparation, J.F.R.-L., G.C., E.C.-C. and J.F.A.-P.; writing—review and editing, J.F.R.-L., G.C., E.C.-C. and J.F.A.-P.; visualization, J.F.R.-L., J.F.A.-P. and E.C.-C.; supervision, G.C., E.C.-C. and J.F.A.-P. All authors have read and agreed to the published version of the manuscript.

**Funding:** This work was funded by a Consejo Nacional de Ciencia y Tecnología—CONACYT.

**Institutional Review Board Statement:** Not applicable since no patients have been involved.

**Informed Consent Statement:** Informed consent was obtained from all subjects involved in the study.

**Data Availability Statement:** The data presented in this study are available on request at the discretion of the corresponding author.

**Acknowledgments:** The first author would like to acknowledge CONACYT for the financial support during the master's program.

**Conflicts of Interest:** The authors declare no conflict of interest.

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
