# Peer review of "Experimental Characterization of A-AFiM, an Adaptable Assistive Device for Finger Motions"

_machines, doi:10.3390/machines10040280_

Round 1

Reviewer 1 Report

I reviewed this paper for journal Sensors before. I saw the authors have modified the manuscript based on my suggestions.

Author Response

Thanks for this valuable comment. English has been carefully revised as suggested.

Reviewer 2 Report

I have found few mistypes, there are probably more, co I suggest to ask for a proofreading. The mitypes are:

  • fin-ger (finger)
  • Hiper-extension (Hyper-extension)

I am not sure if the tenses are appropriately used.

Author Response

(The authors gave the same response as above.)

Reviewer 3 Report

The authors have made significant improvements to the paper since the previous version. They have satisfactorily addressed the questions of the reviewers. It is an interesting work that will be of interest to readers.

Author Response

Thanks for this valuable comment. English has been carefully revised as suggested.

This manuscript is a resubmission of an earlier submission. The following is a list of the peer review reports and author responses from that submission.

Round 1

Reviewer 1 Report

The paper presents an adaptable assistive device for finger rehabilitation. The mechanism to imitate the finger motions is selected to be a slider-crank type which is flexible in the length of a linkage. Motion analysis in terms of trajectory and power consumption is shown in the paper of the mechanism. Different people's samples of the trajectories are recorded to verify that the designed device can cover most of the people's characteristics.

The comments are as follows:

  1. Using a mechanical mechanism to imitate different people's finger trajectories is a quite old technique. Obviously, the problem is that the trajectory of a classical mechanical mechanism, such as the slider-crank one, is fixed after fixing the lengths of the linkages. Although the device proposed in this paper has flexibility, the trajectory generated by a slider-crank mechanism cannot perfectly fit the trajectory generated by a finger which is a three degrees of freedom articulated planar mechanism. Therefore, people cannot comfortably use the device. But for disabled people, maybe that is not a problem since their preferences have been lost. However, I still think using a fully actuated three or two degrees of freedom planar mechanism with advance control techniques is the best choice for the purpose of rehabilitation because such a robot can track any trajectories within its workspace.
  2. There is no related work for the comparison of similar devices. The authors need to add more related works and compare them.
  3. Why do you analyse the power consumption? You just put Fig. 9 there. But, what did you use that do? To select motor powers?
  4. It is not clear how do you adjust the length and the flexible linkage. I think the authors should give a technique to compute the suitable length of the device based on the lengths of a patient's fingers. Otherwise, how do you know how to adjust the length of the device?
  5. The PID is confusing since the authors declare that "that allows for path planning through PID control". I think you only can control the speed of the motor and the speed of the EE point because the trajectory is determined by the lengths of the linkages of the device. In Fig. 11, I also see that only the velocity is changing. The positions are empty.
  6. I highly recommend the authors add a accompany video to 

    validate the authenticity of the experiments. It is already a convention in robotic community.

  7. What is 2-GDL in the paragraph under Fig. 14?
  8. English needs to be checked. For example, errors are: the first sentence of section 2; the sentence under Fig. 7; the sentence under Eq. (9); the caption of Fig. 8.

Author Response

We would like to thank the reviewers for their insightful comments and suggestions. Please, find below in this document our reply to your comments.

Reviewer 1 Report (Round 1)

The paper presents an adaptable assistive device for finger rehabilitation. The mechanism to imitate the finger motions is selected to be a slider-crank type which is flexible in the length of a linkage. Motion analysis in terms of trajectory and power consumption is shown in the paper of the mechanism. Different people's samples of the trajectories are recorded to verify that the designed device can cover most of the people's characteristics.

Comment 1:

Using a mechanical mechanism to imitate different people's finger trajectories is a quite old technique. Obviously, the problem is that the trajectory of a classical mechanical mechanism, such as the slider-crank one, is fixed after fixing the lengths of the linkages. Although the device proposed in this paper has flexibility, the trajectory generated by a slider-crank mechanism cannot perfectly fit the trajectory generated by a finger which is a three degrees of freedom articulated planar mechanism. Therefore, people cannot comfortably use the device. But for disabled people, maybe that is not a problem since their preferences have been lost. However, I still think using a fully actuated three or two degrees of freedom planar mechanism with advance control techniques is the best choice for the purpose of rehabilitation because such a robot can track any trajectories within its workspace.

Reply 1:

Thanks for this valuable comment.

Indeed, the full motion of a finger requires at least three degrees of freedom. However, the specific rehabilitation task does not require to fully replicate all the feasible motions of a finger. Instead, a single repetitive finger motion path is necessary and sufficient to provide proper assisted finger exercising. Accordingly, the proposed one-DOF mechanism is sufficient to fulfil the design requirements with beneficial features also in terms of costs and easiness of use. It is to note also that initial set up adjustments can be made on the link lengths of the proposed mechanism in case the therapist wishes to repetitively exercise the finger along a different path.

The above text has been also added in the revised text at pag 2. Lines 80-87.

Comment 2:

There is no related work for the comparison of similar devices. The authors need to add more related works and compare them.

Reply 2:

Thanks for this valuable comment.

The following references have been added in the revised paper.

[12] Su, Y., Wu, K., Lin, C.H., Yu, Y., & Lan, C. (2018). Design of a Lightweight Forearm Exoskeleton for Fine-Motion Rehabilitation. 2018 IEEE/ASME International Conference on Advanced Intelligent Mechatronics (AIM), 438-443.]

[13] Attal, A., & Dutta, A. (2021). Design of a variable stiffness index finger exoskeleton. Robotica, 1-17. doi:10.1017/S0263574721000965

[14] Li M, He B, Liang Z, Zhao C-G, Chen J, Zhuo Y, Xu G, Xie J and Althoefer K (2019) An Attention-Controlled Hand Exoskeleton for the Rehabilitation of Finger Extension and Flexion Using a Rigid-Soft Combined Mechanism. Front. Neurorobot. 13:34. doi: 10.3389/fnbot.2019.00034

[15] Carbone, G., Ceccarelli, M., Capalbo, C., Caroleo, G., & Morales-Cruz, C. (2021). Numerical and experimental performance estimation for a ExoFing - 2 DOFs finger exoskeleton. Robotica, 1-13. doi:10.1017/S0263574721001375

[16] Dickmann  T, Wilhelm NJ, Glowalla  C, Haddadin  S, van der Smagt  P and Burgkart  R (2021) An Adaptive Mechatronic Exoskeleton for Force-Controlled Finger Rehabilitation. Front. Robot. AI 8:716451. doi: 10.3389/frobt.2021.716451

[17] N. Sun, G. Li and L. Cheng, "Design and Validation of a Self-Aligning Index Finger Exoskeleton for Post-Stroke Rehabilitation," in IEEE Transactions on Neural Systems and Rehabilitation Engineering, vol. 29, pp. 1513-1523, 2021, doi: 10.1109/TNSRE.2021.3097888.

[18] H. Li, L. Cheng, N. Sun and R. Cao, "Design and Control of an Underactuated Finger Exoskeleton for Assisting Activities of Daily Living," in IEEE/ASME Transactions on Mechatronics, doi: 10.1109/TMECH.2021.3120030.

The description of similar devices has been improved by adding several additional references with the following comparisons and discussion that has been added in the revised paper at pag 2. Lines 62-77:

“In reference [12,13] the authors proposed a design of a lightweight forearm exoskeleton for fine-motion rehabilitation using a slider crack and four-bar mechanism with several links and actuators. Other authors as in reference [14] propose a combined rigid-soft mechanism for a hand exoskeleton. In reference [15] a numerical and experimental validation of ExoFing (a “DoF finger mechanism exoskeleton) is investigated by focusing on the kinematic model and numerical simulations. A novel mechatronic exoskeleton for finger rehabilitation with sensors detection motions and control is proposed in [16]. The configuration of this exoskeleton can be fully reconstructed using three angular position sensors. In reference [17] a novel index exoskeleton is proposed with three motors for helping post-stroke patients perform finger and training. Moreover, in reference [18] a novel underactuated finger exoskeleton to assist grasping tasks for the elderly with wear muscles strength is designed, the weight of the wearable part of the proposed exoskeletons is 127g and the overall weight is 476g. However, the existing solutions have some limitations in terms of complexity of the control architecture as well as in terms of user-friendliness, which can be addressed with the proposed design solution.

Comment 3:

Why do you analyse the power consumption? You just put Fig. 9 there. But, what did you use that do? To select motor powers?

Reply 3:

Thanks for your comment.

In the new version of the manuscript, the following comments have been added at pag 8. Lines 238-246:

“A power consumption and motor torque analysis has been carried out to analyze the expected performance of the proposed device as well as to properly select the motors for the proposed A-AFiM mechanism. For this purpose, the operation of the proposed device has been considered during flexion and extension exercising. The proposed simulation considers the average force of a human finger of Ff = 1N, Ff = 3N and Ff =5N [15-18] has been applied to the EE. The simulation also considers the contribution of gravity. In Figure 9(a), a picture of the maximum power consumption is shown, whereas in Figure 9(b) a maximum motor torque of the A-AFiM device is shown, including the proposed force in EE”.

Comment 4:

It is not clear how do you adjust the length and the flexible linkage. I think the authors should give a technique to compute the suitable length of the device based on the lengths of a patient's fingers. Otherwise, how do you know how to adjust the length of the device?

Reply 4:

Thanks for your comment. The following text has been added to clarify this issue at pag 5. Lines 170-179.

“The length of the crank link is half of straight length of the flexo-extension rotated path on the ?-axis. Then, changes in the length of the crank link can modify the path amplitude of point E of the mechanism. The design of a variable C is proposed in Figure 5, whose length  between its external nodes A and B varies in proportion of length  [26]. The C is composed of two links connected by a passive rotational joint, point F, and a screw, whose end is attached to G point of link 5 through a ball joint, allowing rotations only. The other point of contact between the screw and link 4 is achieved through a nut; that   varies when screw turns. Similar triangles are formed by points EFG and AFB and they modify the angle ? as reported in [27]”.

Figure 5. The variable length crank C

Comment 5:

The PID is confusing since the authors declare that "that allows for path planning through PID control". I think you only can control the speed of the motor and the speed of the EE point because the trajectory is determined by the lengths of the linkages of the device. In Fig. 11, I also see that only the velocity is changing. The positions are empty.

Reply 5:

Thanks for this valuable comment.

The following text has been added in the revised paper to clarify this issue at pag 10. Lines 273-277:

“A simple close loop PID control is implemented to regulate the motor speed by using the feedback of an incremental encoder. The motor speed is an important parameter has it directly affect the speed of the finger exercising. Accordingly, this parameter can be monitored in real time. The user interface, Fig.11, includes a control for changing the motor speed according to the user needs.”.

Figure 11. The developed graphical user interface

Comment 6:

I highly recommend the authors add a accompany video to validate the authenticity of the experiments. It is already a convention in robotic community.

Reply 6:

Thanks for your comment. A video is available online video at:

https://www.youtube.com/watch?v=HIPcus1tLO0

Comment 7:

What is 2-GDL in the paragraph under Fig. 14?

Reply 7:

Thanks for your comment. We have corrected this typo and changed 2-GDL as 2-DoF, in the revised text.

Comment 8:

English needs to be checked. For example, errors are: the first sentence of section 2; the sentence under Fig. 7; the sentence under Eq. (9); the caption of Fig. 8.

Reply 8:

Thanks for your comment. English has been carefully revised as suggested.

Reviewer 2 Report

The paper presents a rehabilitation robot design for stroke patients. The work is interesting. The authors have developed and evaluated a prototype. The paper lacks mainly in the presentation making it hard to fully understand the author's contributions.

  • The background of the work - specifically, the type of rehabilitation that is targeted is not clear until Section 2, where the authors mention finger rehabilitation after stroke. The authors are recommended to add a paragraph or two to the Introduction to provide more information about the same. 
  • The same applies to the specific contributions made by this paper. It is not until much later in the paper that it becomes clear.
  • The authors have not mentioned what A-Afim stands for in the paper. One can guess from the title, but the authors must clarify in the text.

Author Response

We would like to thank the reviewers for their insightful comments and suggestions. Please, find below in this document our reply to your comments.

Reviewer 2 Report (Round 1)

The paper presents a rehabilitation robot design for stroke patients. The work is interesting. The authors have developed and evaluated a prototype.

Comment 1:

The paper lacks mainly in the presentation making it hard to fully understand the author's contributions.

Reply 1:

Thanks for your comment. The contribution, advantages, and limitations of the proposed device has been improved by adding the following description at pag 2. Lines 87-95.

The proposed design has innovative aspects in terms of low-cost and user-friendly features. It includes a novel end-effector that allows easy adaptation and user comfort, Moreover, the specific design prevents risks of impacts between the device and the finger that are beneficial from safety viewpoint. The proposed novel device can be easily adjusted to different patients and different exercising protocols. The portability of the device allows for self-treatment at home. The new end-effector, user-friendly interface, and the easy-configuration capabilities allow patients to use the proposed device autonomously from home, thus enabling rehabilitation training without requiring continuous assistance by a physiotherapist.  

Comment 2:

The background of the work - specifically, the type of rehabilitation that is targeted is not clear until Section 2, where the authors mention finger rehabilitation after stroke. The authors are recommended to add a paragraph or two to the Introduction to provide more information about the same.

Reply 2:

Thanks for your comment.

The description of similar devices has been improved by adding several additional references with the following comparisons and discussion that has been added in the revised paper at pag 2. Lines 62-77:

 “In reference [12,13] the authors proposed a design of a lightweight forearm exoskeleton for fine-motion rehabilitation using a slider crack and four-bar mechanism with several links and actuators. Other authors as in reference [14] propose a combined rigid-soft mechanism for a hand exoskeleton. In reference [15] a numerical and experimental validation of ExoFing (a “DoF finger mechanism exoskeleton) is investigated by focusing on the kinematic model and numerical simulations. A novel mechatronic exoskeleton for finger rehabilitation with sensors detection motions and control is proposed in [16]. The configuration of this exoskeleton can be fully reconstructed using three angular position sensors. In reference [17] a novel index exoskeleton is proposed with three motors for helping post-stroke patients perform finger and training. Moreover, in reference [18] a novel underactuated finger exoskeleton to assist grasping tasks for the elderly with wear muscles strength is designed, the weight of the wearable part of the proposed exoskeletons is 127g and the overall weight is 476g. However, the existing solutions have some limitations in terms of complexity of the control architecture as well as in terms of user-friendliness, which can be addressed with the proposed design solution.

The following references have been added in the revised paper.

[12] Su, Y., Wu, K., Lin, C.H., Yu, Y., & Lan, C. (2018). Design of a Lightweight Forearm Exoskeleton for Fine-Motion Rehabilitation. 2018 IEEE/ASME International Conference on Advanced Intelligent Mechatronics (AIM), 438-443.]

[13] Attal, A., & Dutta, A. (2021). Design of a variable stiffness index finger exoskeleton. Robotica, 1-17. doi:10.1017/S0263574721000965

[14] Li M, He B, Liang Z, Zhao C-G, Chen J, Zhuo Y, Xu G, Xie J and Althoefer K (2019) An Attention-Controlled Hand Exoskeleton for the Rehabilitation of Finger Extension and Flexion Using a Rigid-Soft Combined Mechanism. Front. Neurorobot. 13:34. doi: 10.3389/fnbot.2019.00034

[15] Carbone, G., Ceccarelli, M., Capalbo, C., Caroleo, G., & Morales-Cruz, C. (2021). Numerical and experimental performance estimation for a ExoFing - 2 DOFs finger exoskeleton. Robotica, 1-13. doi:10.1017/S0263574721001375

[16] Dickmann  T, Wilhelm NJ, Glowalla  C, Haddadin  S, van der Smagt  P and Burgkart  R (2021) An Adaptive Mechatronic Exoskeleton for Force-Controlled Finger Rehabilitation. Front. Robot. AI 8:716451. doi: 10.3389/frobt.2021.716451

[17] N. Sun, G. Li and L. Cheng, "Design and Validation of a Self-Aligning Index Finger Exoskeleton for Post-Stroke Rehabilitation," in IEEE Transactions on Neural Systems and Rehabilitation Engineering, vol. 29, pp. 1513-1523, 2021, doi: 10.1109/TNSRE.2021.3097888.

[18] H. Li, L. Cheng, N. Sun and R. Cao, "Design and Control of an Underactuated Finger Exoskeleton for Assisting Activities of Daily Living," in IEEE/ASME Transactions on Mechatronics, doi: 10.1109/TMECH.2021.3120030.

Comment 3:

The same applies to the specific contributions made by this paper. It is not until much later in the paper that it becomes clear.

Reply 3:

Thanks for your comment. The contribution, advantages, and limitations of the proposed device has been improved by adding the following description at pag 2. Lines 91-95.

The proposed design has innovative aspects in terms of low-cost and user-friendly features. It includes a novel end-effector that allows easy adaptation and user comfort, Moreover, the specific design prevents risks of impacts between the device and the finger that are beneficial from safety viewpoint. The proposed novel device can be easily adjusted to different patients and different exercising protocols. The portability of the device allows also self treatment at home. The new end-effector, user-friendly interface, and the easy-configuration capabilities allow patients to use the proposed device autonomously from home, thus enabling rehabilitation training without requiring continuous assistance by a physiotherapist. 

Comment 4:

The authors have not mentioned what A-Afim stands for in the paper. One can guess from the title, but the authors must clarify in the text.

Reply 4:

Thanks for your comment. In section 1 at pag 2. Lines 78-80 the name of the mechanism is introduced as:

 “In order to support the rehabilitation therapies for fingers during flexion and extension motion, this paper proposed an experimental characterization of A-AFiM (Adaptable Assistive Device for Finger Motions)”.
